# Indirect effects of the early phase of the COVID-19 pandemic on the coverage of essential maternal and newborn health services in a rural subdistrict in Bangladesh: results from a cross-sectional household survey

Shema Mhajabin ,[1] Aniqa Tasnim Hossain,[1] Nowrin Nusrat,[1] Sabrina Jabeen,[1] Shafiqul Ameen,[1] Goutom Banik,[1] Tazeen Tahsina,[1] Anisuddin Ahmed,[1] Qazi Sadeq-ur Rahman,[1] Emily S Gurley,[2] Sanwarul Bari,[1] Atique Iqbal Chowdhury,[1] Shams El Arifeen,[1] Rajesh Mehta,[3] Ahmed Ehsanur Rahman [1,4]

SEA, RM and AER are joint senior authors.

For numbered affiliations see end of article.

**Correspondence to**
Dr Ahmed Ehsanur Rahman;
ehsanur@icddrb.org

## ABSTRACT

**Objective** This paper presents the effect of the early phase of COVID-19 on the coverage of essential maternal and newborn health (MNH) services in a rural subdistrict of Bangladesh.

**Design** Cross-sectional household survey with random sampling.

**Setting** Baliakandi subdistrict, Rajbari district, Bangladesh.

**Participants** Data were collected from women who were on the third trimester of pregnancy during the early phase of the pandemic (111) and pre-pandemic periods (115) to measure antenatal care (ANC) service coverage. To measure birth, postnatal care (PNC) and essential newborn care (ENC), data were collected from women who had a history of delivery during the early phase of the pandemic (163) and pre-pandemic periods (166).

**Exposure** Early phase of the pandemic included a strict national lockdown between April and June 2020, and pre-pandemic was defined as August–October 2019.

**Outcome of interest** Changes in the coverage of selected MNH services (ANC, birth, PNC, ENC) during the early phase of COVID-19 pandemic compared with the pre-pandemic period, estimated by two-sample proportion tests.

**Findings** Among women who were on the third trimester of pregnancy during the early phase of the pandemic period, 77% (95% CI: 70% to 85%) received at least one ANC from a medically trained provider (MTP) during the third trimester, compared with 83% (95% CI: 76% to 90%) during the pre-pandemic period (p=0.33). Among women who gave birth during the early phase of the pandemic period, 72% (95% CI: 66% to 79%) were attended by an MTP, compared with 63% (95% CI: 56% to 71%) during the pre-pandemic period (p=0.08). Early initiation of breast feeding was practised among 38% (95% CI: 31% to 46%) of the babies born during the early phase of the pandemic period. It was 37% (95% CI: 29% to 44%) during the pre-pandemic period (p=0.81). The coverage of ANC, birth, PNC and ENC did not differ by months

of pandemic and pre-pandemic periods; only the coverage of at least one ANC from an MTP significantly differed among the women who were 7 months pregnant during the early phase of the pandemic (35%, 95% CI: 26% to 44%) and pre-pandemic (49%, 95% CI: 39% to 58%) (p=0.04).

**Conclusion** The effect of the early phase of the pandemic including lockdown on the selected MNH service coverage was null in the study area. The nature of the lockdown, the availability and accessibility of private sector health services in that area, and the combating strategies at the rural level made it possible for the women to avail the required MNH services.

## Strengths and limitations of this study

► This paper provides population-level service coverage during the early phase of the pandemic, which national-level routine information systems cannot capture.

► Additionally, this paper compares the service coverage of the essential maternal and newborn health services before and during the pandemic to understand the effect of the pandemic in rural areas where usually service coverage is less compared with urban areas.

► This study acknowledges the limitation of the cross-sectional design in inferring any causality and estimating the impact.

► This study adapted the questionnaire based on the standard Bangladesh Demographic Health Survey, Bangladesh Maternal Mortality Survey and Multiple Indicator Cluster Survey tools, which are validated and globally accepted.

► This study considered only 6 months of recall for assessing the coverage during the early phase of the pandemic period.

## INTRODUCTION

COVID-19, caused by SARS-CoV-2, has caused more than 242 million confirmed cases across 219 countries and territories and has taken around 5 million lives as of October 2021, making it the deadliest pandemic since the Spanish flu in 1918.[1 2] However, this is only the tip of the iceberg. Any pandemic negatively impacts the social, economic, and political landscape globally, regionally, and nationally.[3–6] The unexpected health burden compels the policymakers to shift focus from routine healthcare and repurpose the existing resources to tackle the more ominous threats posed by the pandemic.[7–9] It is, therefore, likely that the morbidity and mortality due to other causes may arise during a pandemic, especially in countries with limited resources.[10 11]

The risks of COVID-19 infection and its adverse clinical consequences are relatively low among children and the younger population.[12 13] A multicountry cohort study found that pregnant individuals with COVID-19 diagnosis had higher rates of adverse outcomes, including maternal mortality, pre-eclampsia and preterm birth compared with pregnant individuals without COVID-19 diagnosis.[14] Mothers and newborns are also vulnerable to the indirect effects of the pandemic.[15–18] Focusing on the emergency response may result in the deprioritisation of essential health services, including maternal and newborn health (MNH). The availability of essential MNH services may become severely compromised due to the repurposing of limited resources, overwhelming demand of other emergency services, healthcare workers' absence and deaths, and reduced service hours or closure of health facilities.[15 19–21] Similarly, the access to and utilisation of essential MNH services may be adversely affected by the hesitancy to seek care from health facilities due to the fear of contracting the virus and lack of reliable transportation due to the restrictive movement measures such as the lockdown.[22 23] A systematic review and meta-analysis of 40 studies reported that the global burden of maternal deaths, stillbirth and maternal depression has increased during the COVID-19 pandemic, with a high disparity between the high-income and low/middle-income country settings.[24] Another study reported that the under-5 deaths increased by 45% per month and the maternal deaths have increased by 36% per month, across 118 countries due to the indirect effect of the COVID-19 pandemic using the survey data.[15] According to a United Nations report, COVID-19 may have contributed to an estimated 11 000 additional maternal deaths and 228 000 additional child deaths in South Asia in 2020 using routine health information data such as the District Health Information System (DHIS2).[16] Such adverse indirect effects of the COVID-19 pandemic may abrade the major gains in improving MNH achieved over the past few decades.

On 8 March 2020, Bangladesh declared the first confirmed case of COVID-19, and the first death was reported on 18 March 2020. Since then, Bangladesh has reported more than 770 000 confirmed cases, including more than 11 000 deaths.[25] The Government of Bangladesh declared 'general leave' for a week on 26 March 2020 for interrupting the viral transmission chain and flattening the epidemic curve comparable with lockdown.[26] Later, the 'general leave' was extended for 2 months. In the beginning, all businesses except medical services were closed, and all kinds of movement except medical emergencies were restricted. The lockdown measures were attenuated in the second month of lockdown. A sharp decline was observed in the utilisation of essential MNH services during the lockdown period (March–May 2020) in Bangladesh using the DHIS2.[27 28] For example, the study reported that there was a 41%, 52% and 56% decrease in April 2020 compared with January 2020 for institutional normal deliveries in upazila health complex (UHC), district hospital and tertiary hospitals. However, this was based on the DHIS2, a routine health information system, which collects utilisation data from public health facilities and some selected private facilities. Therefore, this might not reflect the change in the coverage of essential MNH services estimated with a true population-level denominator. This paper presents the effect of the early phase of the pandemic on the coverage of essential MNH services in Baliakandi, a rural subdistrict in Bangladesh. This paper will further observe the differences in selected MNH indicators by month of the early phase of the pandemic.

## METHODS

### Study design

We conducted a cross-sectional household survey to estimate the coverage of essential MNH services during the early phase of the pandemic and pre-pandemic periods. Antenatal care (ANC) from a medically trained provider (MTP), birth attended by an MTP, essential newborn care (ENC) and postnatal care (PNC) from an MTP within 72 hours of birth were identified as tracers for the essential MNH services. As the local administration had implemented pandemic measures, the April–June 2020 period was operationally defined as the early phase of the pandemic period. For identifying the pre-pandemic period (routine circumstances), we considered the following factors. China reported the first COVID-19 case in December 2019, and several global and national measures were taken between January and March 2020 and there may have been an impact on normal life and routine services. Before this, November 2019–January 2020 period is the winter season in Bangladesh, and various cultural programmes and family events are organised in this season, possibly leading to changes in the service use pattern. Hence, we considered August–October 2019 as the pre-pandemic period for comparison purposes.

### Study setting

We conducted the study in a rural subdistrict named Baliakandi, under the Rajbari district in Bangladesh.

Since September 2017, the International Centre for Diarrhoeal Disease Research, Bangladesh (icddr,b) has been conducting health and demographic surveillance in this area.[29] The surveillance covers 261 villages and a population of about 227 540 (in 2019). Through two monthly household visits, the surveillance data collectors collect information on births, deaths, marriages, divorces and migrations. In addition to these vital events, they also collect information on pregnancy and birth outcomes.

### Study participants and outcomes of interest
We used the icddr,b's Baliakandi surveillance data to generate the sampling frame based on specific inclusion criteria. Then, we used simple random sampling to select the required number of participants from each category (online supplemental figure 1). The women were randomly selected if they were into the following four groups:

1. Women who were on the third trimester of pregnancy during April–June 2020 and gave birth in July 2020: 126 women were randomly selected to assess the effect of the early phase of a pandemic on ANC.
2. Women who were on the third trimester of pregnancy during August–October 2019 and gave birth in November 2019: 126 women were randomly selected to assess the effect of the pre-pandemic on ANC.
3. Women who gave birth during April–June 2020: 178 women were randomly selected to assess the effect of the early phase of a pandemic on birth, PNC and ENC.
4. Women who gave birth in August–October 2019: 178 women were randomly selected to assess the effect of the pre-pandemic on birth, PNC and ENC.

We calculated the sample size to detect a minimum effect size of 33 percentage points (relative) between the early phase of the pandemic and the pre-pandemic periods at 80% power (two sided) and 5% error probability. We used the coverage estimates (rural) reported in 2017–2018 Bangladesh Demographic and Health Survey (BDHS) for the selected essential MNH services as proxies for the pre-pandemic status.[30] The coverage estimates (rural) for the selected MNH services for the pandemic period were collected using DHIS2 data. The unadjusted sample size was adjusted for a non-response rate of 10% and the clustering effect with a 1.10 design effect. Details regarding the sample size calculation for each of the essential MNH services are presented in online supplemental table 1.

### Data collection tools
We used an interviewer-administered structured questionnaire to interview the eligible respondents. The questions were primarily adopted from the BDHS, Bangladesh Maternal Mortality Survey and Multiple Indicator Cluster Survey.[30–32] The questionnaire consisted of seven main sections: household information; socioeconomic status; respondent's background; pregnancy, birth and PNC; ENC; care-seeking during emergencies; telemedicine service; and COVID-19-related knowledge and

experiences. We translated the questionnaire into Bangla and pretested it before finalisation.

### Training and data collection
We recruited 15 data collectors and 2 field supervisors from the local communities who were familiar with the local context, culture and dialect. The data collection team received 5 days of face-to-face training, which was followed by 2 days of field testing. The training was conducted by the study investigators and other master trainers with special expertise and experience in household surveys.

Data were collected between 7 November and 25 December 2020. The field supervisors revisited a random sample of 5% of households to ensure quality. In addition, the field supervisors conducted unscheduled visits to observe the interviews and monitor the quality of data collection. Weekly meetings were organised to solve any emerging issues related to data collection. All questionnaires were manually reviewed by the field supervisors and the field research managers before sending for data entry.

We maintained social distancing and followed the national infection prevention and control guidelines during training and data collection.[33] All interviews were conducted outdoors. The data collection team received appropriate personal protective equipment such as masks, gloves, hand sanitisers, aprons and face shields. Also, we distributed reusable face masks to the respondents to wear during the interviews.

### Data analysis plan
We used the statistical software package STATA V.14.2 (StataCorp, College Station, Texas, USA) for data analysis.[34] At first, descriptive statistics were used to report the background characteristics of the respondents. We used the principal component analysis to generate an asset score for each household.[35] The asset scores were used to rank the households into quintiles. We reported the coverage of selected essential MNH services for the early phase of the pandemic and pre-pandemic periods separately with 95% CIs. Additionally, the service coverage of selected MNH indicators was presented by months of the early phase of the pandemic and pre-pandemic with 95% CI. The operational definitions of each of the indicators are presented in online supplemental table 2, and the definitions of providers are presented in online supplemental table 3. Two sample proportion tests were conducted to explore the difference in the coverage of selected essential MNH services between the early phase of the pandemic and pre-pandemic periods. We reported no significant difference at $p < 0.05$.

### Patient and public involvement
There was no patient or public involvement in the study.

## RESULTS
We interviewed 115 women (non-response rate 9%) who were on the third trimester of pregnancy during the

**Table 1** Background characteristics of the respondents; presented in column percentage

| Background characteristics | Women who were on the third trimester of pregnancy | | Women who had a history of birth outcome | |
| --- | --- | --- | --- | --- |
| | Pre-pandemic | Early phase of pandemic | Pre-pandemic | Early phase of pandemic |
| | N=115 | N=111 | N=166 | N=163 |
| | % | % | % | % |
| **Age** | | | | |
| 14 and below | 0 | 1 | 0 | 0 |
| 15–19 | 23 | 23 | 14 | 22 |
| 20–24 | 31 | 36 | 33 | 37 |
| 25–29 | 21 | 25 | 26 | 22 |
| 30–34 | 19 | 13 | 22 | 12 |
| 35+ | 6 | 2 | 5 | 7 |
| **Education** | | | | |
| No education | 3 | 1 | 5 | 3 |
| Primary incomplete* | 9 | 10 | 9 | 11 |
| Primary complete† | 7 | 9 | 12 | 11 |
| Secondary incomplete‡ | 56 | 48 | 52 | 52 |
| Secondary complete or higher§ | 25 | 32 | 22 | 23 |
| **Religion** | | | | |
| Muslim | 83 | 88 | 89 | 91 |
| Hinduism/Buddhism/Christianity | 17 | 12 | 11 | 9 |
| **Parity** | | | | |
| Primiparous | 41 | 41 | 32 | 44 |
| Multiparous | 59 | 59 | 68 | 56 |
| **Occupation** | | | | |
| Housewife | 96 | 95 | 95 | 90 |
| Engaged in any income-generating activity | 4 | 5 | 5 | 10 |
| **Wealth quintile** | | | | |
| Lowest | 22 | 14 | 15 | 15 |
| Second | 23 | 19 | 24 | 24 |
| Middle | 25 | 23 | 25 | 23 |
| Fourth | 13 | 23 | 18 | 23 |
| Highest | 17 | 21 | 18 | 15 |
| Total | 100 | 100 | 100 | 100 |

*1–4 years of schooling.
†5 years of schooling.
‡6–9 years of schooling.
§10 or more years of schooling.

pre-pandemic period and 111 women (non-response rate 12%) who were on the third trimester of pregnancy during the early phase of the pandemic period. We also interviewed 166 women (non-response rate 7%) who gave birth during the pre-pandemic period and 163 women (non-response rate 9%) who gave birth during the early phase of the pandemic period. Table 1 describes the background characteristics of each group of respondents. Approximately one-third of the women were aged 20–24 years. Around half of them had incomplete secondary education (6–9 years of schooling), and

one-fourth had complete secondary or higher education (10 or more years of schooling). Most of the respondents were Muslim. Approximately 40% of the women were primigravidae (first-time pregnant). Less than 10% of them were engaged in any income-generating activities. No notable difference was observed in characteristics across the groups of respondents.

Figure 1 summarises the coverage of essential MNH services during pre-pandemic and early phase of pandemic periods. Among women who were on the third trimester of pregnancy during the early phase of the

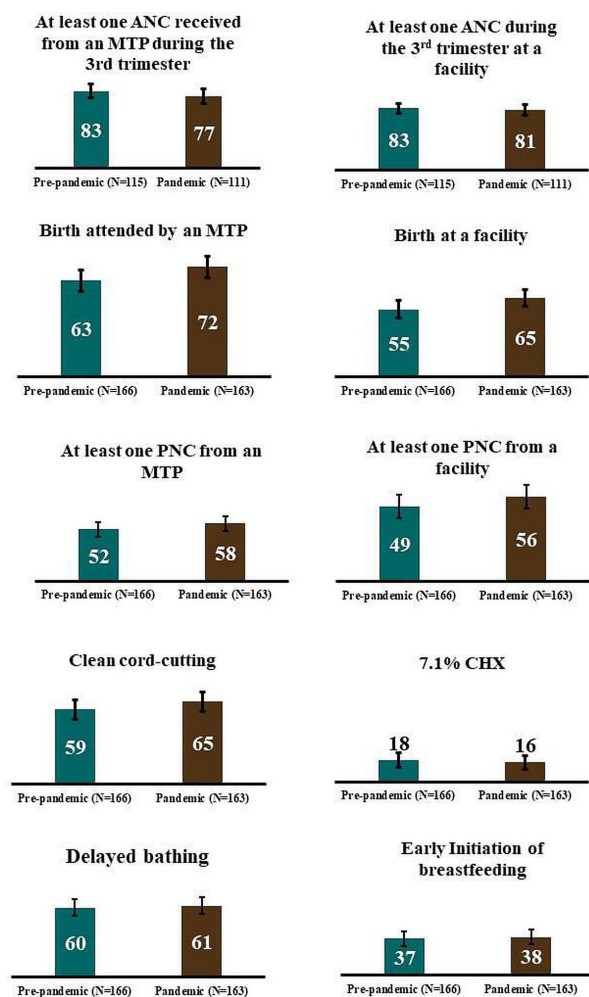

**Figure 1** Coverage of essential MNH services during the pre-pandemic and the pandemic periods, presented in percentage with 95% CIs. ANC, antenatal care; CHX, chlorhexidine; MNH, maternal and newborn health; MTP, medically trained provider; PNC, postnatal care.

pandemic period, 77% (95% CI: 70% to 85%) received at least one ANC from an MTP during the third trimester, which was 83% (95% CI: 76% to 90%) during the pre-pandemic period (p=0.33). Similarly, among women who were on the third trimester of pregnancy during the early phase of the pandemic period, 81% (95% CI: 74%

to 88%) received at least one ANC from a health facility during the third trimester, which was 83% (95% CI: 76% to 90%) during the pre-pandemic period (p=0.77).

Among women who gave birth during the pandemic period, 72% (95% CI: 66% to 79%) were attended by an MTP, whereas only 63% (95% CI: 56% to 71%) of women were attended by an MTP during the pre-pandemic period (p=0.08). The coverage of facility birth was 65% (95% CI: 58% to 72%) during the pandemic period and 55% (95% CI: 47% to 62%) during the pre-pandemic period (p=0.06). The coverage of at least one PNC from an MTP was 58% (95% CI: 51% to 60%) during the pandemic period and 52% (95% CI: 45% to 60%) during the pre-pandemic period (p=0.28).

Among babies born during the pandemic period, the coverage of clean cord cutting practice was 65% (95% CI: 58% to 72%), whereas only 59% (95% CI: 52% to 67%) of babies born during the pre-pandemic period had clean cord cutting (p=0.26). The 7.1% chlorhexidine was applied to 16% (95% CI: 10% to 22%) of babies born during the pandemic period, which was 18% (95% CI: 12% to 24%) during the pre-pandemic period (p=0.61). Early initiation of breast feeding was practised among 38% (95% CI: 31% to 46%) of the babies born during the pandemic period, whereas it was 37% (95% CI: 29% to 44%) during the pre-pandemic period (p=0.81).

Figure 2 illustrates the coverage of ANC received in the third trimester of pregnancy by months during the pandemic and pre-pandemic periods. During the pandemic period, only 35% (95% CI: 26% to 44%) of women received at least one ANC from an MTP on their 7th month of pregnancy (April 2020), whereas it was 49% (95% CI: 39% to 58%) for pre-pandemic (August 2019) (p=0.04). Similarly, only 36% (95% CI: 27% to 45%) of women received at least one ANC from a health facility, and 32% (95% CI: 23% to 40%) of women received at least one ANC from a private health facility during the 7th month of pregnancy (April 2020). However, these estimates were significantly lower than that of the pre-pandemic period (August 2019) (p=0.05). No notable difference was observed in the ANC coverage during the 8th and 9th months of pregnancy between the pandemic and pre-pandemic periods (p>0.05).

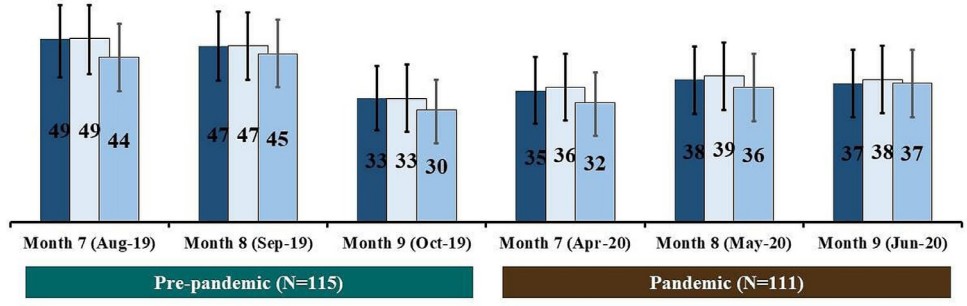

**Figure 2** Coverage of ANC received in the third trimester during the pre-pandemic and the pandemic periods; presented in percentages by months of pregnancy. ANC, antenatal care; MTP, medically trained provider.

 

**Table 2** Type of provider from whom ANC was received in the third trimester during the pre-pandemic and early phase of pandemic periods; presented in percentages

| Type of providers | Pre-pandemic (95% CI) | Early phase of pandemic (95% CI) | P value |
|---|---|---|---|
| Doctor | 80 (73 to 87) | 77 (70 to 78) | 0.64 |
| Nurse/midwife/paramedic | 20 (13 to 27) | 11 (5 to 17) | 0.06 |
| FWV | 11 (6 to 17) | 7 (2 to 12) | 0.29 |
| CSBA | 0 | 0 | |
| SACMO/MA/CHCP/HA/FWA | 8 (3 to 13) | 13 (6 to 19) | 0.24 |
| NGO worker/TBA | 2 (0 to 4) | 6 (2 to 11) | 0.08 |

Multiple responses considered.
ANC, antenatal care; CHCP, community healthcare provider; CSBA, community skilled birth attendant; FWA, family welfare assistant; FWV, family welfare visitor; HA, health assistant; MA, medical assistant; NGO, Non-government Organization; SACMO, subassistant community medical officer; TBA, traditional birth attendant.

Among the women who were on the third trimester of pregnancy, 23% did not receive any ANC from an MTP during the pandemic period, whereas only 17% did not receive any ANC from an MTP during the pre-pandemic period (p=0.33) (online supplemental figure 2). During the pandemic period, only 43% of women on the third trimester of pregnancy received one ANC from an MTP, but during the pre-pandemic, it was only 40% (p=0.62) We did not observe any notable difference (statistically significant at p<0.05) in the above-mentioned distribution of ANC status between the pandemic and the pre-pandemic periods. The care-seeking for ANC using telemedicine services was very low and not different (p=0.99) between pandemic and pre-pandemic periods (online supplemental figure 3).

Table 2 presents the type of healthcare provider from whom the women received ANC during their third trimester of pregnancy. Among the women who were on the third trimester of pregnancy during the pandemic period, 11% received ANC from a nurse/midwife/

paramedic, whereas 20% received ANC from these providers during the pre-pandemic period (p=0.06). No notable difference (statistically significant at p<0.05) was observed in the above-mentioned distribution of ANC providers between the pandemic and the pre-pandemic periods.

Figure 3 presents the coverage of birth attended by an MTP, birth in a health facility, birth in a private health facility and birth through C-sections. Among women who gave birth during the pandemic period, the coverage of birth attended by an MTP was around 70% across all 3 months. Similarly, the coverage of facility birth was more than 60% across all 3 months. Around 50% of births occurred in a private health facility, and the rates were consistent across the 3 months. The coverage of birth attended by an MTP was 73% (95% CI: 59% to 85%) in April 2020, whereas it was only 64% (95% CI: 46% to 79%) in August 2019 (p=0.46). We did not observe any notable difference (statistically significant at p<0.05) in the coverage of birth attended by an MTP, birth in

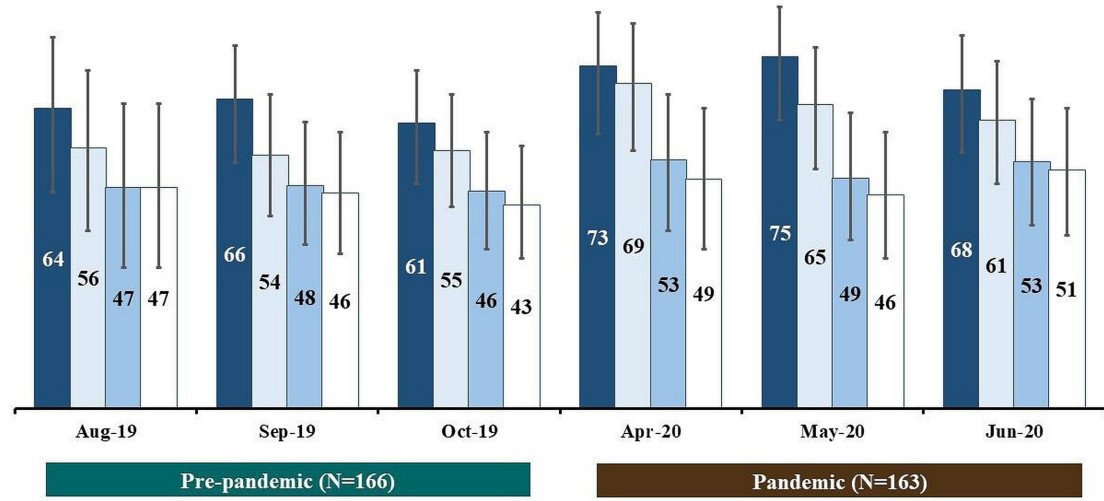

**Figure 3** Coverage of births attended by a medically trained provider (MTP), birth in a health facility, birth in a private health facility, and birth through C-sections during the pre-pandemic and pandemic periods; presented in percentages by calendar months.

**Table 3** Type of provider attending births during the pre-pandemic and early phase of pandemic periods; presented in percentages

| Type of providers | Pre-pandemic (95% CI) | Early phase of pandemic (95% CI) | P value |
|---|---|---|---|
| Doctor | 49 (41 to 56) | 53 (46 to 61) | 0.41 |
| Nurse/midwife /paramedic | 62 (55 to 69) | 71 (64 to 78) | 0.08 |
| FWV | 0 | 1 (0 to 2) | 0.31 |
| CSBA | 2 (−1 to 4) | 0 | 0.32 |
| SACMO/MA/CHCP /HA/FWA | 4 (1 to 6) | 2 (0 to 4) | 0.32 |
| NGO worker/TBA | 40 (33 to 48) | 35 (28 to 41) | 0.31 |

CHCP, community healthcare provider; CSBA, community skilled birth attendant; FWA, family welfare assistant; FWV, family welfare visitor; HA, health assistant; MA, medical assistant; SACMO, subassistant community medical officer; TBA, traditional birth attendant.

a health facility, birth in a private health facility, and birth through C-sections during the pandemic and pre-pandemic periods for any of the reporting months.

Table 3 presents the type of healthcare providers attending the births which were conducted during the pre-pandemic and the pandemic periods. Among the women who gave birth during the pandemic period, 53% were attended by a doctor and 71% by a nurse/midwife/paramedic. Among the women who gave birth during the pre-pandemic period, 49% were attended by a doctor and 62% by a nurse/midwife/paramedic. No notable difference (statistically significant at p<0.05) was observed in the distribution of providers attending births between the pandemic and the pre-pandemic periods.

Figure 4 shows the coverage of PNC from an MTP, PNC from a health facility, and PNC from a private health facility during the pre-pandemic and the pandemic periods. Among women who gave birth during the pandemic period, the coverage of PNC by an MTP within

72 hours of birth was around 60% across all 3 months. Similarly, the coverage of PNC in a health facility was around 55% across all 3 months. The coverage of PNC by an MTP was 57% (95% CI: 42% to 71%) in April 2020, whereas it was only 53% (95% CI: 35% to 70%) in August 2019 (p=0.79). We did not observe any notable difference (statistically significant at p<0.05) in the coverage of PNC by an MTP, PNC in a health facility, and PNC in a private health facility during the pandemic and pre-pandemic periods for any of the reporting months.

Figure 5 presents the coverage of different components of ENC among the newborns during the pre-pandemic and the pandemic periods. Among babies born at the early phase of the pandemic (April 2020), the coverage of early initiation of breast feeding was 24% (95% CI: 13% to 39%), which was much lower than that of the remaining months (May–June 2020) during the pandemic period and the pre-pandemic period (p=0.37). No notable difference (statistically significant at p<0.05) was observed in the coverage of other components of ENC between the pandemic and the pre-pandemic periods by calendar months.

## DISCUSSION

The healthcare-seeking experience for ANC, birth, PNC and ENC during the early phase of the COVID-19 pandemic in rural Bangladesh was not found affected in this study. We found that the coverage of selective essential MNH services during the early phase of the pandemic was not different from that of the pre-pandemic period in this rural subdistrict in Bangladesh. However, national-level data from the routine health system of Bangladesh (DHIS2) showed a substantial decline in the utilisation of essential MNH services.[27] The possible explanations for such contradicting findings are summarised under two broad themes, that is, how the service utilisation was restored (health system response and community coping

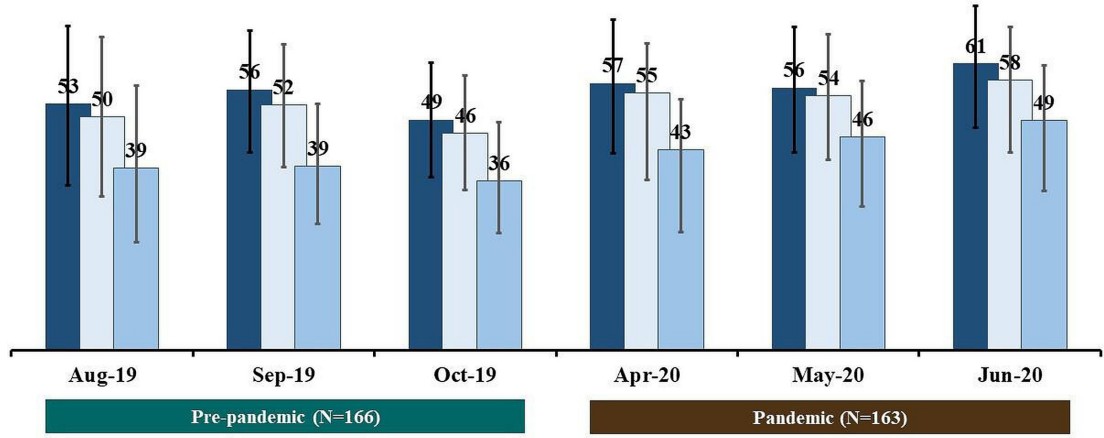

**Figure 4** Coverage of PNC by a medically trained provider (MTP), PNC in a health facility, PNC in a private health facility among births that occurred during the pre-pandemic and pandemic periods; presented in percentages by calendar months. PNC, postnatal care.

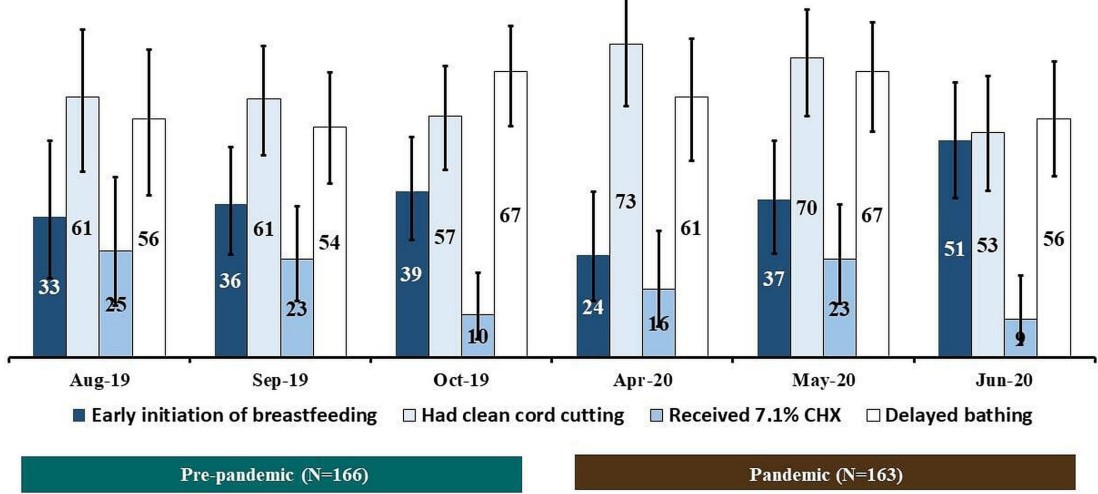

**Figure 5** Coverage of different components of ENC during the pre-pandemic and pandemic periods; presented in percentages by calendar months. CHX, chlorhexidine; ENC, essential newborn care.

strategy and resilience) and why we did not see any difference in the service utilisation (private sector coverage, enforcement of lockdown measures and measurement issues in the routine health information system) are discussed below.

### Health systems responses

Several health systems initiatives taken by the Government of Bangladesh might have contributed to overcoming the initial shock of the pandemic and preventing a substantial decline in the coverage of key MNH services. In March 2020, the government developed the Bangladesh Preparedness and Response Plan for COVID-19, which also emphasised the importance of maintaining essential health and nutrition services during the pandemic.[36] After the initial focus on COVID-19-specific preparation and responses, several initiatives were taken to promote the availability and readiness of essential MNH services. In May 2020, the Ministry of Health, with technical inputs from the development partners and professional societies, developed the 'National Guideline for providing essential Maternal, Newborn and Child Health Services in the context of COVID-19'. The prime objective of this guideline was to inform the healthcare providers and facility-level and district-level managers of the importance of providing essential MNH services in the context of COVID-19.[37] It also provided specific guidance on continuing the essential services safely by maintaining physical distancing, using personal protective equipment, and promoting infection prevention and control practices. The Ministry of Health organised a series of online sessions to sensitise the facility-level and district-level managers. Afterward, face-to-face training was organised for the frontline healthcare workers responsible for providing MNH services. Then national-level policymakers organised virtual meetings with facility-level, district-level and regional-level managers to review the MNH utilisation data in respective health facilities and discuss strategies for overcoming the bottlenecks, barriers

and operational challenges. Special initiatives were also taken for establishing functional triage, isolation unit for patients with suspected COVID-19 and red zone for patients with confirmed COVID-19 at referral facilities, which promoted a safer environment at the facility for receiving routine services and might have contributed to restoring the public confidence for availing essential MNH services. In June 2020, the Government of Bangladesh recruited and deployed 2000 doctors and 5000 nurses for COVID-19 response.[38] The influx of additional human resources helped redeploy the MNH services to their original duties and minimise the impact of COVID-19 on essential services. These national efforts might have positively influenced minimising the disruption in the availability and provision of essential services in our study settings. According to the national routine health information system, there was little difference in the utilisation of essential MNH services from public health facilities in the Baliakandi subdistrict during the early phase of the pandemic period.[27] This is particularly applicable for the UHC, which is the primary reference point for all health centres in the study site. We also did not observe any notable difference in the utilisation of MNH services from the district hospital during that period, which is the primary reference point for all the UHCs in that district.[27] It indicates that most of the public health referral facilities continued providing routine services despite prioritising the COVID-19-specific services. Since the referral facilities have more human resources than health centres, which have only one to two dedicated providers, it was easier for them to adopt an alternate approach for continuing essential MNH services.

### Community coping strategy and resilience

Bangladesh is one of the most vulnerable countries to climate variability and fluctuation.[39 40] A quarter of the land in Bangladesh is barely above sea level and is susceptible to the effects of seawater rise and increased salinity due to climate change. Bangladesh also experiences

other forms of natural disasters like floods and cyclones quite frequently.[41] The people in Bangladesh have learnt to cope with the emergencies and show resilience by focusing on rebuilding after the initial shock, mostly through local-level initiatives.[42 43] Awareness regarding the importance of receiving MNH services from appropriate places and providers has increased substantially and consistently over the past couple of decades, even in the rural context. Hence, the awareness and the resilient nature of local communities might have contributed to finding ingenious ways and coping strategies to avail the essential MNH service during the pandemic, overcoming the challenges imposed by the lockdown.

## Private sector coverage

Bangladesh has a pluralistic health system constituted by the public sector, for-profit formal private sector and informal care providers.[44 45] Although the Government of Bangladesh has a large network of health infrastructure throughout the country, the majority of MNH services are provided by the poorly regulated for-profit private sector and unregulated informal care providers. According to the 2017–2018 BDHS, more than two-thirds of the ANC and facility births took place in private health facilities.[30] In the past decade, the coverage of births in private health facilities has increased by fourfold.[30 46] A similar trend was also observed for other essential MNH services such as ANC, PNC and ENC. Although almost all the public health facilities report to DHIS2, very few private hospitals and private practitioners report to DHIS2. This disconnect between the larger market share of the private sector and disproportionately minor contribution in the DHIS2-based reporting system may explain our study findings. We did not observe any notable change in the coverage of essential MNH services during the early phase of the pandemic period. In contrast, the DHIS2-based reporting system was indicating a substantial decline in service utilisation nationally. In our study site, around 55% of births took place in the health facilities during the pre-pandemic period, of which about 80% were in private health facilities. The condition did not change during the pandemic period as 65% of births took place in health facilities, of which more than 80% were in private health facilities. We observed similar patterns for ANC and PNC services. In the rural subdistrict where we conducted the study, there are three for-profit private health facilities that provide MNH services.[29] In addition, there are 16 for-profit private facilities and 2 non-profit NGO clinics in the district headquarters, and many of them offer MNH services. Therefore, the prioritisation of emergency services related to COVID-19 exclusively through the public health facilities may explain the initial decline in utilisation of essential MNH services in those facilities, but the overwhelming majority in the market share of the private sector and the continuation of the routine services may have contributed to sustaining the essential MNH services in the early stage of the pandemic.

## Enforcement of lockdown measures

The Government of Bangladesh imposed a nationwide restriction of movements in late March 2020 to interrupt the pathways of COVID-19 transmission.[26 47 48] However, the government declared it as 'general leave' instead of lockdown, which undermined the importance of adhering to the recommended measures among the mass population. At the beginning of the lockdown, all the government and non-government offices, factories and academic institutes were asked to shut down. A large proportion of city dwellers travelled back to their village, maintaining very little social distancing and other infection prevention and control measures.[47] After a couple of weeks, some government offices and factories, including all ready-made garment factories, were allowed to resume their operations, forcing many people to return to their workplace. Hence, the restrictions regarding interdistrict and within-district movements gradually weaned off, and the public transportation options slowly resumed back with some infection prevention and control regulations.[26] Moreover, the lockdown measures were less coercively imposed in rural settings as the transportation services were reasonably available, allowing people to move within the subdistrict and to the district headquarters when required. In our study, less than 5% of the respondents mentioned lack of transportation as a barrier to care-seeking, and less than 10% reported spending extra money for the vehicle rent while receiving essential MNH services during the lockdown period. Based on a national survey, around two-thirds of the pregnant women in rural settings can reach a private health facility or provider within 5 km of travel distance.[30] It implies that people could avail the essential MNH services from nearby private health facilities when the services were temporarily disrupted in public health facilities in the early stage of the pandemic.

## Measurement issues

As mentioned earlier, there are limitations in estimating the coverage of health services through the routine health information system in Bangladesh. The Government of Bangladesh has made a substantial investment in strengthening the routine health information system by introducing DHIS2 in 2009.[44] All the public health facilities and a few private health facilities report their health service use data to DHIS2. Although MNH is a priority in the DHIS2-based reporting system in Bangladesh, the system's reach is not enough to accurately capture population-level denominators such as live births, which are essential for measuring the coverage of MNH services. In 2019, DHIS2 reported a total of 931 723 live births in Bangladesh.[27] Based on the crude birth rate reported in 2017–2018 BDHS, the size of the annual birth cohort is more than 3 million in Bangladesh.[30] Similarly, DHIS2 reported that around 655 542 women received four ANC services from the reporting facilities in 2019.[27] However, an estimated 1 410 000 (47% of 3 million births) women received four or more ANC services based on the national survey estimate.[30] Moreover, there are issues with the accuracy and quality of reporting

in the routine health information systems, including DHIS2, particularly in low/middle-income country settings.[49–52] Due to these gaps in accurately capturing the denominators and numerators through the DHIS2-based reporting system, the decline observed in the utilisation of essential MNH services in public health facilities in the early stage of the pandemic may not reflect the true change in the coverage of relevant services.

Second, there were gaps and delays in the documentation and reporting of routine services in the initial phase of the pandemic, particularly during the lockdown period due to the reshaping of the healthcare providers' responsibilities and health systems responses. The Government of Bangladesh prepared the National Preparedness and Response Plan for the pandemic and repurposed the existing infrastructure, human resources, and logistics to prevent and treat COVID-19.[36] Initially, triage was introduced in all public health facilities, and some hospitals were dedicated to treating COVID-19. Additionally, a COVID-19 dashboard has been developed and is updated daily.[25] This portal displays data on the number of laboratories testing for COVID-19 and the number of health facilities treating patients with COVID-19. The dashboard also displays information on the number of tests performed, number of cases identified, number of people in quarantine and isolation, and number of deaths daily. It also provides an overview of the stock availability of COVID-19-related logistics of all health facilities. The frontline healthcare workers, facility-level and district-level managers, and national policymakers prioritised capturing and reporting of COVID-19-related data to monitor the relevant implementation and response status. Hence, the introduction and emphasis on COVID-19-related services, documentation, and reporting might have overwhelmed the healthcare providers and the health systems resulting in the initial gaps and delays of reporting other routine services, including MNH. For example, we accessed DHIS2 in June 2020 and observed a sharp decline in the utilisation of essential MNH services from public health facilities in March–May 2020.[27] We accessed DHIS2 in September 2020 and found that the service utilisation numbers were updated for the same time (March–May 2020), resulting in a much smaller decline than the pre-COVID-19 period. Therefore, the true impact on the coverage of essential MNH services might be less than what was originally observed based on DHIS2 utilisation data in the early phase of the pandemic.

## CONCLUSION

Our study findings reported no significant changes in the overall coverage of MNH services in the rural area of Bangladesh. Moreover, the changes were not even significant for monthly comparison during the early phase of the pandemic and pre-pandemic periods. It highlights the importance of expanding and reinforcing the routine health information system to cover the information from the private health facilities and community services for comprehensive monitoring of essential services, particularly during the emergency response. The results also reveal the strength of local-level coping strategies to overcome the immediate shock.

**Author affiliations**
[1]Maternal and Child Health Division (MCHD), International Centre for Diarrhoeal Disease Research Bangladesh, Dhaka, Bangladesh
[2]Johns Hopkins University, Baltimore, Maryland, USA
[3]WHO Regional Office for South-East Asia, New Delhi, Delhi, India
[4]Usher Institute, University of Edinburgh, Edinburgh, UK

**Acknowledgements** The authors would like to acknowledge the kind support of the Baliakandi demographic surveillance site which is supported by the CHAMPS Project of the Gates Foundation in Emory University throughout the data collection. icddr,b is also grateful to the Governments of Bangladesh, Canada, Sweden and the UK for providing unrestricted/institutional support.

**Contributors** SM and AER are the co-principal investigators of this study, and they conceptualised the study. SM and AER are guarantors of this study. SM took the lead in developing the first draft of the manuscript and prepared all tables and figures. SJ, SA, GB, TT, AA, ESG, SB, and AIC provided their critical feedback on each draft of the manuscript. SM, AH, NN have conducted the data analysis and interpreted the findings with continuous advice from QS-uR and AER. All the authors have contributed significantly to designing the study conducted and preparing the final draft of the manuscript. All authors have read and approved the final version of the manuscript. AER, SEA, and RM have led the process as the senior authors.

**Funding** The study was supported by the WHO under the grant of BGT-0031/2020.

**Competing interests** None declared.

**Patient consent for publication** Not required.

**Ethics approval** This study involves human participants and icddr,b Institutional Review Board has approved this study (PR-20085). Written informed consent was sought from each study participant before the interview. In the case of respondents who cannot read or write, we obtained audio-recorded verbal informed consent.

**Provenance and peer review** Not commissioned; externally peer reviewed.

**Data availability statement** Data are available upon reasonable request. The data will be available upon request as per the organisations' policy.

**ORCID iDs**
Shema Mhajabin http://orcid.org/0000-0001-7023-015X
Ahmed Ehsanur Rahman http://orcid.org/0000-0001-9216-1079

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
