## [Reviewer comments · BMJ Open]

ARTICLE DETAILS

TITLE (PROVISIONAL)	Indirect effects of the early phase of the COVID-19 pandemic on the coverage of essential maternal and newborn health services in a rural sub-district in Bangladesh: results from a cross-sectional household survey
AUTHORS	Mhajabin, Shema; Hossain, Aniq; Nusrat, Nowrin; Jabeen, Sabrina; Ameen, Shafiqul; Banik, Goutom; Tahsina, Tazeen; Ahmed, Anisuddin; Sadeq-ur Rahman, Qazi; Gurley, Emily; Bari, Sanwarul; Chowdhury, Atique; Arifeen, Shams E.; Mehta, Rajesh; Rahman, Ahmed

VERSION 1 – REVIEW

REVIEWER	Jayakody, Hemali Family Health Bureau, Ministry of Health, Sri Lanka
REVIEW RETURNED	26-Sep-2021

GENERAL COMMENTS	General comments: Authors have assessed a very important research question in the present pandemic in the SEARO. A good effort. But there were typos and grammatical errors which need further revision. The manuscript could have been more focused and simplified. In the middle of the manuscript, the authors have mixed the findings with available DHIS 2 data which makes the reader /reviewer confused about the situation. Since this is not directly related to the study, suggest removing that part. Suggest removing the multiple non-familiar abbreviations for the international reader. Abstract: Findings section doesn't answer the study question. It needs findings of the hypothesis tested in the study (although it is a negative finding). The conclusion doesn't match the study question. Introduction: Line 45 to 55 - information not related to the study question. Methods: Two study populations need to be clearly defined. It is not clear the relationship between the surveillance program mentioned and the sample selection. Give more details on the sampling method. Is it cluster sampling? (mentioned in the sample size calculation). I suggest including that in a figure. Data analysis: Authors have mentioned multiple composite indices in the introduction and rest of the methods section. Also, they have presented many figures on the individual variable comparison. This section should depict actually what was analyzed in the study.
--

	Results: In general, authors should not explain all findings in the text. Identify key findings and present that in the text. Recommend to use more tables than figures as the reader might get confused with some information presented. The results section lacks focus on the research question. Figure 2- suggest removing as the findings do not add any additional information to the study question. Discussion: A minimum effort was taken to explain the findings - contrary to the widely accepted norm, lockdown did not impact the service coverage in MNH. Either the arguments are unrelated or too dilute and the reader feels lost. Suggest revising the discussion section completely. The authors have not discussed the limitations of the study including the very long recall bias and non-validating the information with other sources (eg: medical records of mothers).
--	--

REVIEWER	Kazmi, Kescha Hospital For Sick Children, Pediatrics
REVIEW RETURNED	16-Oct-2021

GENERAL COMMENTS	Major Comments This manuscript describes the indirect effects of COVID-19 lockdown measures on the coverage of essential maternal and newborn health services in Baliakandi, a rural sub-district in Bangladesh. Authors utilized a cross-sectional household survey to estimate coverage during lockdown and pre-lockdown periods. Interestingly, no significant differences in the overall coverage of MNH services were found during lockdown and pre-lockdown periods. The authors provided compelling reasons for their findings, including community resilience and measurement issues. Only minor issues identified. The manuscript is worthy of publication. Specific Comments Introduction Page 6 Could shorten the first paragraph. Where COVID-19 was first reported is common knowledge now. Line 31 – would caution against saying women of reproductive ages are at lower risk for adverse clinical consequences from COVID-19 infection. Infected pregnant women are at higher risk for poor outcomes (JAMA Pediatr. 2021;175(8):817-826. doi:10.1001/jamapediatrics.2021.1050)
---

Page 7

Line 27 – Briefly describe in a sentence or two what the national lockdown consisted of in Bangladesh as lockdown measures differed from country to country. Were there internal travel restrictions, cessation of public transport, school closures, etc.? I know it's discussed in the discussion but a brief explanation in the introduction provides context for the reader at the start.

Line 38 – Should write out what these abbreviations stand for initially (UHC, DH, DHIS2)

Methods

Page 8

Line 45- Should write out what stand icddr,b stands for initially

Page 9

Line 23 – How was the minimum effect size of 33% estimated?

Results

Page 13

Line 43 – CI should have 95% in front

Table 1 – Some of the percentages in the chart do not add up to 100%

- Women who were third trimester pregnant, lockdown, age category percentages add up to 101%
- Women who were third trimester pregnant, pre-lockdown, education, percentages add up to 99%
- Women who were third trimester pregnant, pre-lockdown, parity, percentages add up to 99%

Discussion

Any studies in LMICs that similarly showed maintenance of MNH services during early phase of the COVID-19 pandemic?

Page 18

	Line 27 – “In May 2202” needs to be corrected, I assume 2020 Page 19 Line 8 – “In June 2010” I assume this is supposed to be 2020 as well Page 20 Line 50 – I wonder about the wording of “ineffective lockdown”. What is an effective lockdown? Perhaps “enforcement of lockdown measures” might be more appropriate? References References need to be in BMJ referencing style (https://authors.bmj.com/writing-and-formatting/formatting-your-paper/) Supplementary Table 2 Indicator 5 Was it supposed to be “proportion of women giving birth” instead of “Proportion of newborn giving birth”
--	--

VERSION 1 – AUTHOR RESPONSE

Reviewer 1:

1. General comments: Authors have assessed a very important research question in the present pandemic in the SEARO. A good effort.

Response: Thank you for your encouraging comment.

2. But there were typos and grammatical errors which need further revision. The manuscript could have been more focused and simplified.

Response: Thank you. We have reviewed the manuscript and made necessary changes.

3. In the middle of the manuscript, the authors have mixed the findings with available DHIS 2 data which makes the reader /reviewer confused about the situation. Since this is not directly related to the study, suggest removing that part.

Response: In the early phase of COVID-19 pandemic, service utilization of essential health services was observed and interpreted using routine health information system such as DHIS2. There are studies which have measured the additional deaths due to decreased in service utilization using DHIS2 data. We have mentioned in the introduction regarding the use of DHIS2 data globally (page 5) and in the context of Bangladesh (page 6). At that time, no studies assessed service coverage at the

population level which we tried to measure in this study. Therefore, we have addressed the difference of service coverage found in DHIS2 and at the population level.

To address your comment, we are elaborating more regarding DHIS2 data and its importance in this study in the introduction (page 5).

4. Suggest removing the multiple non-familiar abbreviations for the international reader.

Response: Thank you. We have elaborated some abbreviations which are important for the country context and removed the non-familiar abbreviations for the global readers on page 5.

5. Abstract: Findings section doesn't answer the study question. It needs findings of the hypothesis tested in the study (although it is a negative finding). The conclusion doesn't match the study question.

Response: Thank you. We have revised the findings and conclusion on page 2 and 3.

6. Introduction: Line 45 to 55 - information not related to the study question.

Response: Thank you. The study question was to identify the service coverage of selected MNH services during lockdown and pre-lockdown period. Line 45-55 said, "demand of other emergency services, healthcare workers' absence and deaths and reduced service hours or closure of health facilities (14, 18-20). Similarly, the access to and utilization of essential MNH services may be adversely affected by the hesitancy to seek care from health facilities due to the fear of contracting the virus and lack of reliable transportation due to the restrictive movement measures such as the lockdown (21, 22)".

Here, we tried to explain the possible reasons of decreased service coverage during the early phase of the pandemic such as lockdown. This study aimed to understand the effect of the early phase of pandemic on service coverage. However, to address your comment, we are decreasing the amount of information in these lines.

7. Methods: Two study populations need to be clearly defined. It is not clear the relationship between the surveillance program mentioned and the sample selection. Give more details on the sampling method. Is it cluster sampling? (mentioned in the sample size calculation). I suggest including that in a figure.

Response: Thank you. We have added a supplementary figure for sampling.

8. Data analysis: Authors have mentioned multiple composite indices in the introduction and rest of the methods section. Also, they have presented many figures on the individual variable comparison. This section should depict actually what was analyzed in the study.

Response: Thank you. We have mentioned the selected maternal and newborn indicators in the method section and detailed definition is presented in supplementary table 2. On page 9, we have explained that the coverage of each selected indicator during early phase of pandemic and pre-pandemic with confidence interval were explored. Also, we conducted two sample proportion test two identify the differences in coverage of the selected maternal and newborn indicators from the two periods (early phase of the pandemic and pre-pandemic).

9. Results: In general, authors should not explain all findings in the text. Identify key findings and present that in the text. Recommend to use more tables than figures as the reader might get confused with some information presented. The results section lacks focus on the research question.

Response: Thank you. We have reduced the text in the findings section. We have converted Figure 3 and Figure 5 in table format.

The tables and figures clearly depict the results of the research question of this study. Table 1 provided the background characteristics of the study participants. Figure 1 summarized the coverage of selected indicators during early phase of pandemic and pre-pandemic which was the main objective of this study to understand the effect of early phase of pandemic on selected indicators by comparing the coverage to the pre-pandemic period. Then, figure 2-5 have tried to show any differences of the indicators between pandemic and pre-pandemic period that could exist by months. Table 2 and table 3 have presented information on type of providers who provided antenatal care services and delivery services during the pandemic and pre-pandemic period.

However, based on your suggestion we are rephrasing the study objective to align it with the findings section.

10. Figure 2- suggest removing as the findings do not add any additional information to the study question.

Response: Thank you for your comment. This figure provides an added and important information to figure 1 which mentioned that there was not any difference in overall ANC coverage among the third trimester pregnant women of lockdown and pre-lockdown period. Figure 2 presented that the coverage of at least one ANC from a medically trained provider differed significantly among the women who were seven months pregnant during pandemic and pre-pandemic. Moreover, this also serve the study objective. Therefore, we would like to keep this figure.

11. Discussion: A minimum effort was taken to explain the findings - contrary to the widely accepted norm, lockdown did not impact the service coverage in MNH. Either the arguments are unrelated or too dilute and the reader feels lost. Suggest revising the discussion section completely.

Response: Thank you. We have appropriately given arguments in the discussion to explain the findings of this study. In the beginning of discussion, we reported no substantial difference regarding the true population coverage of selected MNH services between pandemic and pre-pandemic period which is contradictory to the accepted norm. We explained the role of health system response and local level resilience to present how we responded to the pandemic at national and local level. Then, we laid out our discussion explaining why we did not see any difference or decline in the health service utilization. The possible explanations were the service coverage by private health facilities, measurement issues of routine health data, and enforcement of lockdown.

Under private service coverage, we explained routine health system of Bangladesh, DHIS2, do not have data from the private health facilities, but they are the prime provider of maternal and newborn health services, according to national surveys. The larger share of private market is not captured by DHIS2 which explains our findings.

Under measurement issues of DHIS2, we discussed the gaps and delays in the documentation and reporting of routine services in the initial phase of the pandemic, particularly during the lockdown period due to the reshaping of the health care providers' responsibilities and health systems responses. These gaps and delays could have an effect on the observed coverage of essential MNH services in DHIS2 during early phase of pandemic which in reality was lesser.

However, based on your comment we have reorganized the discussion points so that the readers can follow it better.

12. The authors have not discussed the limitations of the study including the very long recall bias and non-validating the information with other sources (eg: medical records of mothers).

Response: Thank you for raising it. We have listed the recall issues under the strength and limitations section on page 3. In this study, the recall period was of nine months which is less than the recall period (three years) used in standard surveys such as DHS, MICS, and etc. To minimize the recall errors, this study consulted and adapted questions from Bangladesh Demographic Health Survey (BDHS), Bangladesh Maternal Mortality Survey (BMMS), and Multiple Indicator Cluster Surveys (MICS). We also recruited the data collectors from the local communities, organized face-to-face training and conducted household visits for interviewing the women for improving the quality of the data.

Reviewer 2:

1. This manuscript describes the indirect effects of COVID-19 lockdown measures on the coverage of essential maternal and newborn health services in Baliakandi, a rural sub-district in Bangladesh. Authors utilized a cross-sectional household survey to estimate coverage during lockdown and pre-lockdown periods. Interestingly, no significant differences in the overall coverage of MNH services were found during lockdown and pre-lockdown periods. The authors provided compelling reasons for their findings, including community resilience and measurement issues. Only minor issues identified. The manuscript is worthy of publication.

Response: Thank you for your valuable comment.

2. Introduction: Page 6-Could shorten the first paragraph. Where COVID-19 was first reported is common knowledge now.

Response: Thank you. We have revised it on page 4.

3. Line 31 – would caution against saying women of reproductive ages are at lower risk for adverse clinical consequences from COVID-19 infection. Infected pregnant women are at higher risk for poor outcomes (JAMA Pediatr. 2021;175(8):817-826. doi:10.1001/jamapediatrics.2021.1050)

Response: Thank you for suggesting it. We have revised the line on page 4.

4. Page 7: Line 27 – Briefly describe in a sentence or two what the national lockdown consisted of in Bangladesh as lockdown measures differed from country to country. Were there internal travel restrictions, cessation of public transport, school closures, etc.? I know it's discussed in the discussion but a brief explanation in the introduction provides context for the reader at the start.

Response: Thank you. We have added few lines regarding national lockdown in Bangladesh on page 5.

5. Line 38 – Should write out what these abbreviations stand for initially (UHC, DH, DHIS2)

Response: We are sorry for the unintentional mistake. We have revised it on page 5.

6. Methods: Page 8, Line 45- Should write out what stand icddr,b stands for initially.

Response: Thank you. We have revised it on page 6.

7. Methods: Page 9, Line 23 – How was the minimum effect size of 33% estimated?

Response: Thank you. On page 7, we have mentioned that the coverage estimates for pre-pandemic was identified from Bangladesh Demographic and Health Survey (BDHS) 2017-18. In supplementary table 1, we have presented the coverage estimates for pandemic using District health information system (DHIS2) data. We are adding this information on page 7.

8. Results: Page 13, Line 43 – CI should have 95% in front.

Response: Thank you. We have revised it.

9. Table 1 – Some of the percentages in the chart do not add up to 100%

- Women who were third trimester pregnant, lockdown, age category percentages add up to 101%
- Women who were third trimester pregnant, pre-lockdown, education, percentages add up to 99%
- Women who were third trimester pregnant, pre-lockdown, parity, percentages add up to 99%

Response: Thank you. We have revised it on page 10.

10. Discussion: Any studies in LMICs that similarly showed maintenance of MNH services during early phase of the COVID-19 pandemic?

Response: Thank you. We did not find any similar study at this stage which was about assessing service utilization at the household level.

11. Page 18: Line 27 – “In May 2202” needs to be corrected, I assume 2020

Response: We are sorry for the typo. We have revised it on page 16.

12. Page 19: Line 8 – “In June 2010” I assume this is supposed to be 2020 as well

Response: We are very sorry for this mistake. We have revised it on page 17.

13. Page 20: Line 50 – I wonder about the wording of “ineffective lockdown”. What is an effective lockdown? Perhaps “enforcement of lockdown measures” might be more appropriate?

Response: We agree with you. We have revised it on page 18.

14. References: References need to be in BMJ referencing style

(https://linkprotect.cudasvc.com/url?a=https%3a%2f%2fauthors.bmj.com%2fwriting-and-formatting%2fformating-your-paper%2f&c=E,1,Xub6s0nNSuhd0mk96RRHHKt71-zb9ziTRxtroA5DafwjIMp7HjV-OZTdpCTNkGjZv5HcTKSSfseL2L4FyvUHPT_TFK4PeQ3GCScofdmDCy20D3TITw,,&typo=1)

Response: We have updated the reference style as BMJ referencing style on page 23-25.

15. Supplementary Table 2: Indicator 5 Was it supposed to be “proportion of women giving birth” instead of “Proportion of newborn giving birth”

Response: Sorry for this mistake. We have revised the supplementary table 2.

VERSION 2 – REVIEW

REVIEWER	Kazmi, Kescha Hospital For Sick Children, Pediatrics
REVIEW RETURNED	30-Nov-2021

GENERAL COMMENTS	Great job on the revisions. Only minor comments on the revised manuscript. Minor comments Small grammatical errors throughout Abstract Page 4 Line 48 – would add p value of 0.33 here in brackets “...during the pre-pandemic period (p=0.33)”. Line 53 – same thing, would add p value in brackets (p=0.08). Line 57 – again, add p value in brackets (p=0.81) Line 57 – Remove sentence “After assessing the service coverage of the selected MNH services during the early phase of pandemic and pre-pandemic, no significant differences were observed (i.e. p-value for at least one ANC from MTP equals to 0.33, p-value for birth attended by a MTP equals to 0.08, and p-value for early initiation of breastfeeding equals to 0.81).” Page 5 Line 5 – Can you add proportions and CI to “Only the coverage of at least one ANC from MTP significantly differed among the women who were seven months pregnant during early phase of pandemic (proportion and CI here) and pre-pandemic (proportion and CI here, p-value<0.05)”. Line 13 – do you know for sure that the reason you didn’t see a significant difference between MNH service coverage was because people accessed the private sector? Perhaps soften that statement Methods Page 9 Study participants and outcomes of interest  - Should state that participants were recruited from icddr,b surveillance data and should explicitly state inclusion and exclusion criteria - This statement should be included in the first paragraph so we know sampling method at the
--

	start “We used the surveillance data to generate the sampling frame based on specific inclusion criteria. Then, we used simple random sampling to select the required number of participants from each category” Results Page 13 Line 32 – include p value here “...during the pre-pandemic period (p=0.33).” Line 39 – include p value here “...during the pre-pandemic period (p=0.77).” Line 41 – remove statement in brackets Tables 2 and 3 – below table in a smaller font, write out each abbreviation so it’s easier to understand the table Good discussion!
--	--

VERSION 2 – AUTHOR RESPONSE

Reviewer 2:

1.Minor comments: Small grammatical errors throughout

Response: We are very sorry for the minor grammatical errors. We have reviewed the manuscript carefully and revised all grammatical errors we came across.

2.Abstract: Page 4 - Line 48 – would add a p-value of 0.33 here in brackets “...during the pre-pandemic period (p=0.33)”.

Response: Thank you. We have added it on page 2.

3.Line 53 – the same thing, would add p-value in brackets (p=0.08). Line 57 – again, add p-value in brackets (p=0.81).

Response: Thank you. We have added it on page 2.

4.Line 57 – Remove sentence “After assessing the service coverage of the selected MNH services during the early phase of pandemic and pre-pandemic, no significant differences were observed (i.e. p-value for at least one ANC from MTP equals 0.33, the p-value for birth attended by an MTP equals to 0.08, and p-value for early initiation of breastfeeding equals to 0.81).”

Response: We have removed this sentence.

5. Page 5, Line 5 – Can you add proportions and CI to “Only the coverage of at least one ANC from MTP significantly differed among the women who were seven months pregnant during the early phase of pandemic (proportion and CI here) and pre-pandemic (proportion and CI here, p -value <0.05)”.

Response: Thank you. We have revised it.

6. Line 13 – do you know for sure that the reason you didn’t see a significant difference between MNH service coverage was that people accessed the private sector? Perhaps soften that statement.

Response: Thank you. We agree with your comment and have paraphrased the statement.

7. Methods: Page 9, Study participants and outcomes of interest, Should state that participants were recruited from icddr,b surveillance data and should explicitly state inclusion and exclusion criteria.

Response: Thank you. We have added a sentence to explain that the women were recruited from icddr,b’s surveillance data. We also have revised the first paragraph of study participants and outcome of interest to clearly state the inclusion and exclusion criteria.

8. This statement should be included in the first paragraph so we know the sampling method at the start “We used the surveillance data to generate the sampling frame based on specific inclusion criteria. Then, we used simple random sampling to select the required number of participants from each category”.

Response: Thank you. We have added this sentence at the beginning of the “study participants and outcome of interest” section.

9. Results: Page 13, Line 32 – include p-value here “...during the pre-pandemic period ($p=0.33$).”

Response: Thank you. We have added the p-value.

10. Results: Line 39 – include p-value here “...during the pre-pandemic period ($p=0.77$).”

Response: Thank you. We have added the p-value.

11. Results: Line 41 – remove statement in brackets

Response: Thank you. We have removed it.

12. Results: Tables 2 and 3 – below table in a smaller font, write out each abbreviation so it’s easier to understand the table

Response: Thank you. We have added it.

13. Good discussion!

Response: Thank you for your comment.